# MEASURING AND MITIGATING INTERFERENCE IN REINFORCEMENT LEARNING

## ABSTRACT

Catastrophic interference is common in many network-based learning systems, and many proposals exist for mitigating it. But, before we overcome interference we must understand it better. In this work, we first provide a definition and novel measure of interference for value-based control methods such as Fitted Q Iteration and DQN. We systematically evaluate our measure of interference, showing that it correlates with forgetting, across a variety of network architectures. Our new interference measure allows us to ask novel scientific questions about commonly used deep learning architectures and develop new learning algorithms. In particular we show that updates on the last layer result in significantly higher interference than updates internal to the network. Lastly, we introduce a novel online-aware representation learning algorithm to minimize interference, and we empirically demonstrate that it improves stability and has lower interference.

## 1 INTRODUCTION

Generalization is a key property of reinforcement learning (RL) algorithms with function approximation. An agent must correctly generalize its recent experience to both states it has not yet encountered and other states it encountered in the past. Generalization has been extensively studied in supervised learning, inputs are sampled iid from a fixed input distribution and the targets are sampled from a fixed conditional distribution.

The distribution of training data is often not iid. When learning from a stream of temporally correlated data, as in RL, the learner might fit the learned function to recent data and potentially overwrite previous learning—for example, the estimated values. This phenomenon is commonly called *interference* or forgetting in RL (Bengio et al., 2020; Goodrich, 2015; Liu et al., 2019; Kirkpatrick et al., 2017; Riemer et al., 2018). The conventional wisdom is that interference is particularly problematic in RL, even single-task RL, because (a) when an agent explores, it processes a sequence of observations, which are likely to be temporally correlated; (b) the agent continually changes its policy, changing the distribution of samples over time; and (c) most algorithms use bootstrap targets (as in temporal difference learning), making the update targets non-stationary.

It is difficult to verify this conventional wisdom, as there is no established online measure of interference for RL. There has been significant progress quantifying interference in supervised learning (Chaudhry et al., 2018; Fort et al., 2019; Kemker et al., 2018; Riemer et al., 2018), with some empirical work even correlating interference and properties of task sequences (Nguyen et al., 2019), and investigations into (un)forgettable examples in classification (Toneva et al., 2019). In RL, recent efforts have focused on generalization and transfer, rather than characterizing or measuring interference. Learning on new environments often results in drops in performance on previously learned environments (Farebrother et al., 2018; Packer et al., 2018; Rajeswaran et al., 2017; Cobbe et al., 2018). DQN-based agents can hit performance plateaus in Atari, presumably due to interference. In fact, if the learning process is segmented in the right way, the interference can be more precisely characterized with TD errors across different game contexts (Fedus et al., 2020). Unfortunately this analysis cannot be done online as learning progresses. Finally, recent work investigated several different possible measures of interference, but did not land on a clear measure (Bengio et al., 2020).

In this paper we advocate for a simpler approach to charactering interference in RL. In most systems the value estimates and actions change on every time-step conflating many different sources of non-stationarity, stochasticity, and error. If an update to the value function interferes, the result of

that updated might not manifest in the policy's performance for several time steps, if at all. Interference classically refers to an update negatively impacting the agent's previous learning—eroding the agent's knowledge stored in the value function. Therefore it makes sense to first characterize interference in the value function updates, instead of the policy or return. We define interference in terms of prediction error for two common approximate dynamic programming algorithms, approximate policy iteration and fitted Q iteration. Most value-based deep RL algorithms are based on these two algorithms. The interference is defined as the change in prediction errors, which is similar to previous definitions of interference in supervised learning. Additionally, our approach yields an online estimate of interference, which can even be directly optimized.

In this work, we provide a clear justification for the use of differences in squared TD errors as the definition of interference. We highlight the definitions of interference at different granularities, and the utility of considering different statistics to summarize interference within iterations versus over time. We evaluate our interference measure by computing the correlation to a forgetting metric, which reflects instability in control performance. We show that high interference correlates with forgetting, and simultaneously show interference and forgetting properties across a variety of architectures and optimization choices. We then use our measure to highlight that updates to internal layers of the network—the representation—contribute much less to interference than updates on the last layer. This motivates the design of a new algorithm that learns representation online that explicitly minimize interference. We conclude with a demonstration that this algorithm does indeed significantly improve stability and reduce interference.

## 2 PROBLEM FORMULATION AND LEARNING ALGORITHMS

In reinforcement learning (RL), an agent interacts with its environment, receiving observations and selecting actions to maximize a reward signal. We assume the environment can be formalized as a Markov decision process (MDP). An MDP is a tuple $(\mathcal{S}, \mathcal{A}, \mathrm{Pr}, R, \gamma, d_0)$ where $\mathcal{S}$ is a set of states, $\mathcal{A}$ is an set of actions, $\mathrm{Pr} : \mathcal{S} \times \mathcal{A} \times \mathcal{S} \to [0, 1]$ is the transition probability, $R : \mathcal{S} \times \mathcal{A} \times \mathcal{S} \to \mathbb{R}$ is the reward function, $\gamma \in [0, 1]$ a discount factor, and $d_0$ is the initial distribution. The goal of the agent is to find a policy $\pi : \mathcal{S} \times \mathcal{A} \to [0, 1]$ to maximize the expected discounted sum of rewards.

Given a fixed policy $\pi$, the action-value function $Q^\pi : \mathcal{S} \times \mathcal{A} \to \mathbb{R}$ is defined as $Q^\pi(s, a) := \mathbb{E}[\sum_{k=0}^\infty \gamma^k R_{t+k+1} | S_t = s, A_t = a]$, where $R_{t+1} = R(S_t, A_t, S_{t+1})$, $S_{t+1} \sim \mathrm{Pr}(\cdot | S_t, A_t)$, and actions are taken according to policy $\pi$: $A_t \sim \pi(\cdot | S_t)$. Given a policy $\pi$, the value function can be obtained using the Bellman operator for action values $\mathcal{T}^\pi : \mathbb{R}^{|\mathcal{S}| \times |\mathcal{A}|} \to \mathbb{R}^{|\mathcal{S}| \times |\mathcal{A}|}$: $(\mathcal{T}^\pi Q)(s, a) := \sum_{s' \in \mathcal{S}} \mathrm{Pr}(s'|s, a) \left[ R(s, a, s') + \gamma \sum_{a' \in \mathcal{A}} \pi(a'|s') Q(s', a') \right]$. $Q^\pi$ is the unique solution of the Bellman equation $\mathcal{T}^\pi Q = Q$.

The optimal value function $Q^*$ is defined as $Q^*(s, a) := \sup_\pi Q(s, a)$, with $\pi^*$ the policy that is greedy w.r.t. $Q^*$. Similarly, the optimal value function can be obtained using the Bellman optimality operator for action values $\mathcal{T} : \mathbb{R}^{|\mathcal{S}| \times |\mathcal{A}|} \to \mathbb{R}^{|\mathcal{S}| \times |\mathcal{A}|}$: $(\mathcal{T}Q)(s, a) := \sum_{s' \in \mathcal{S}} \mathrm{Pr}(s'|s, a) \left[ R(s, a, s') + \gamma \max_{a' \in \mathcal{A}} Q(s', a') \right]$. $Q^*$ is the unique solution of the Bellman equation $\mathcal{T}Q = Q$. We can use neural networks to learn an approximation $Q_{\boldsymbol{\theta}}$ to the optimal action-value, with parameters $\boldsymbol{\theta}$.

In this work, we restrict our attention to *Iterative Value Estimation* algorithms. These are algorithms where there is an explicit evaluation phase with a fixed policy, where the agent has several steps $T_{\mathrm{eval}}$ to improve its value estimates. Two examples of such algorithms are Approximate Policy Iteration (API) and Fitted Q-Iteration (FQI) (Ernst et al., 2005). In API, for the current policy $\pi_k$, the agent updates its estimate of $Q^{\pi_k}$ by taking $T_{\mathrm{eval}}$ steps in the environment and performing a mini-batch update from a replay buffer on each step using the Sarsa update:

$$\boldsymbol{\theta}_{t+1} \leftarrow \boldsymbol{\theta}_t + \alpha \delta_t \nabla_{\boldsymbol{\theta}_t} Q_{\boldsymbol{\theta}_t}(S_t, A_t) \qquad \text{where } \delta_t := R_{t+1} + \gamma Q_{\boldsymbol{\theta}_t}(S_{t+1}, \pi_k(S_{t+1})) - Q_{\boldsymbol{\theta}_t}(S_t, A_t).$$

In FQI, the policy and targets $Q_k$ are held fixed for $T_{\mathrm{eval}}$ steps, with these fixed targets used as a regression target in the update. Again, a mini-batch update update from a replay buffer is used on each step as above, but with a different $\delta = R_{t+1} + \gamma \max_{a' \in \mathcal{A}} Q_k(S_{t+1}, a') - Q_{\boldsymbol{\theta}_t}(S_t, A_t)$. The procedure for both algorithms is summarized in Algorithm 1.

---

**Algorithm 1** Iterative Value Estimation: A General Framework for API and FQI

---

Initialize weights $\boldsymbol{\theta}_0$. Initialize an empty buffer of size $B$.
**for** $t \leftarrow 0, 1, 2, \ldots$ **do**
    **If** $t \bmod T_{\text{eval}} = 0$ **then** $Q_k \leftarrow Q_{\boldsymbol{\theta}_t}$, update $\pi_k$ to be greedy w.r.t $Q_k$, $b_k$ to be $\epsilon$-greedy
    Choose $a_t \sim b_k(s_t)$, observe $(s_{t+1}, r_{t+1})$, and add the transition to the buffer
    Sample a mini-batch of transitions $B_t$ from the buffer and update the weights:
    $\boldsymbol{\theta}_{t+1} \leftarrow \boldsymbol{\theta}_t + \alpha \frac{1}{|B_t|} \sum_{(s,a,r,s') \in B_t} \delta(\boldsymbol{\theta}_t; s, a, r, s') \nabla_{\boldsymbol{\theta}} Q_{\boldsymbol{\theta}_t}(s, a)$
    where for API: $\delta(\boldsymbol{\theta}_t; s, a, r, s') = r + \gamma Q_{\boldsymbol{\theta}_t}(s', \pi_k(s'))) - Q_{\boldsymbol{\theta}_t}(s, a)$
    and for FQI: $\delta(\boldsymbol{\theta}_t; s, a, r, s') = r + \gamma \max_{a'} Q_k(s', a') - Q_{\boldsymbol{\theta}_t}(s, a)$

---

## 3 DEFINING INTERFERENCE IN VALUE ESTIMATION

In this section, we define interference for Iterative Value Estimation algorithms. Because these methods have an evaluation phase which corresponds to one iteration, we can more clearly define interference within one iteration. We discuss interference at four different levels of granularity.

Within each iteration—in each evaluation phase—we can ask: did the agent's knowledge about its value estimates improve or degrade? The evaluation phase is more similar to a standard prediction problem, where the goal is simply to improve the estimates of the action-values towards a clear target. Let $f^*$ be the target function, which is either $f^*(s, a) = Q^{\pi_k}(s, a)$ for API or $f^*(s, a) = \mathbb{E}[R + \max_{a'} Q_k(S', a') | S = s, A = a]$ for FQI.

**Pointwise Interference** At the most fine-grained, we can ask if an update, going from $\boldsymbol{\theta}_t$ to $\boldsymbol{\theta}_{t+1}$, resulted in interference for a specific point $(s, a)$. The change in accuracy at $s, a$ after an update is

$$\text{Accuracy Change}((s, a), \boldsymbol{\theta}_t, \boldsymbol{\theta}_{t+1}) := (f^*(s, a) - Q_{\boldsymbol{\theta}_{t+1}}(s, a))^2 - (f^*(s, a) - Q_{\boldsymbol{\theta}_t}(s, a))^2$$

where if this number is negative it reflects that accuracy improved. This change resulted in interference if it is positive, and zero interference if it is negative, and so we have

$$\text{Pointwise Interference}((s, a), \boldsymbol{\theta}_t, \boldsymbol{\theta}_{t+1}) := \max\left(\text{Accuracy Change}((s, a), \boldsymbol{\theta}_t, \boldsymbol{\theta}_{t+1}), 0\right).$$

**Update Interference** At a less fine-grained level, we can ask if the update generally improved our accuracy—our knowledge in our value estimates—across points.

$$\text{Update Interference}(\boldsymbol{\theta}_t, \boldsymbol{\theta}_{t+1}) := \max\left(\mathbb{E}_{(S,A) \sim d}\left[\text{Accuracy Change}((S, A), \boldsymbol{\theta}_t, \boldsymbol{\theta}_{t+1})\right], 0\right)$$

where $(s, a)$ are sampled according to distribution $d$, such as from a buffer of collected experience.

Notice that this differs from the *expected Pointwise Interference*. There are settings where they could produce notably different values. For example, an agent could have high positive and negative Accuracy Change that cancel. The Update Interference reflects that, on average, the agent's knowledge has not changed: it improved in some places, and degraded in others. The expected Pointwise Interference, on the other hand, would be high, because for some points interference was high. We focus first on Update Interference, since it is the coarser measure; future work is to look in a more fine-grained way at Pointwise Interference.

Both Pointwise Interference and Update Interference are about one step. At an even higher level, we can then ask how much interference we have across multiple steps, both within an iteration and across multiple iterations. At this higher level, it becomes more sensible to consider upper percentiles, to ask if there was significant interference within an iteration and across iterations. For this we take expectations over only the top $\alpha$ percentage of values. In finance, this is typically called the expected tail loss or conditional value at risk. Previous work in RL (Chan et al., 2020) has used conditional value at risk to measure the long-term risk of RL algorithms.

**Iteration Interference** reflects if there was significant interference in updating during the evaluation phase (an iteration). Even a few update steps having significant interference within an iteration could cause significant instability; an average over the steps might wash out those few significant steps. We therefore define Iteration Interference for iteration k using expectation over the top $10\%$ of values

$$\text{Iteration Interference}(k) := \mathbb{E}[X | X \geq \text{Percentile}_{0.9}(X)] \text{ for } X = \text{Update Interference}(\boldsymbol{\theta}_{T,k}, \boldsymbol{\theta}_{T+1,k})$$

where $T$ is the time step in the iteration $k$, uniformly distributed and $\text{Percentile}_{0.9}(X)$ is the $0.9$-percentile of the distribution of $X$. Other percentiles could be considered, where smaller percentiles average over more values and a percentile of $0.5$ gives the median.

**Interference Across Iterations** reflects if an agent has many iterations with significant interference. Once again, even a few iterations with significant interference could destabilize learning; expectations over tails are more suitable than over all iterations. For iteration $K$ a random variable,

$$\text{Interference Across Iterations} := \mathbb{E}[X | X \geq \text{Percentile}_{0.9}(X)] \text{ for } X = \text{Iteration Interference}(K)$$

These definitions are quite generic, assuming only that we have well-defined targets for the evaluation phase and an algorithm that proceeds in iterations. Though we have only discussed API and FQI, the algorithm DQN also fits well into this class—and so could be analyzed—because the use of target networks mimics FQI. The primary difference is that the policy changes within an iteration. This is not logistically an issue, as updates are off-policy from a replay buffer; but, it does add an additional changing variable. We therefore focus our experiments on API and FQI.

## 4 APPROXIMATING ACCURACY CHANGE AND UPDATE INTERFERENCE

The primary difficulty now is estimating the Accuracy Change, which involves the true values $Q^\pi$ for API and expectations over next state and reward for FQI. With a simulator, these can in fact be estimated. For small experiments, therefore, the exact Accuracy Change could be computed. More generally, the cost is prohibitive, and approximations are needed. In this section, we motivate the use of TD errors as a reasonable approximation.

We focus on finding a useful proxy measure for Update Interference, where the simply need our approximation to be reflective of the average of Accuracy Change, rather than get it correct for a particular point. Our goal is to find a proxy measure that is easy to compute and that at least reflects the same sign as the Update Interference. For FQI, the target in Accuracy Change corresponds to the expected TD error, $\mathbb{E}[\delta | S = s, A = a]$, for the $\delta$ defined for FQI. For API, the expected TD error is the Bellman error: $\mathbb{E}[\delta | S = s, A = a] = \mathcal{T}^\pi Q(s,a) - Q(s,a)$, where $\mathcal{T}^\pi Q(s,a) = \mathbb{E}_\pi[R + \gamma Q(S', A') | S = s, A = a]$. The term $\mathcal{T}^\pi Q(s,a)$ does not match the target $f^*(s,a) = Q^\pi(s,a)$. Fortunately, there is quite a lot of theory showing that the Bellman error provides an upper bound on the value error (Williams, 1993), and further that using Bellman errors is sufficient to obtain performance bounds for API (Munos, 2003; 2007; Farahmand et al., 2010).

Therefore, a unified choice to approximate Accuracy Change for both API and FQI is to approximate the expected TD errors. We can get an unbiased sample of these TD errors, but the square of these TD errors does not correspond to the squared expected TD error (Bellman error). Instead, there is a residual term, that reflects the variance of the targets (Antos et al., 2008)

$$\mathbb{E}[\delta^2 | S = s, A = a] = \mathbb{E}_\pi[\delta | S = s, A = a]^2 + \text{Var}_\pi[R + Q_{\boldsymbol{\theta}}(S', A') | S = a, A = a].$$

When we consider the difference in TD errors, after an update, for $(s, a)$, we get

$$\mathbb{E}[\delta(\theta_{t+1})^2 | S = s, A = a] - \mathbb{E}[\delta(\theta_t)^2 | S = s, A = a]$$
$$= \mathbb{E}_\pi[\delta(\theta_{t+1}) | S = s, A = a]^2 - \mathbb{E}_\pi[\delta(\theta_t) | S = s, A = a]^2$$
$$+ \text{Var}_\pi[R + Q_{\boldsymbol{\theta}_{t+1}}(S', A') | S = a, A = a] - \text{Var}_\pi[R + Q_{\boldsymbol{\theta}_t}(S', A') | S = a, A = a].$$

For a given $(s, a)$, we would not expect the variance of the target to change significantly. When subtracting the squared TD errors, therefore, we expect these residual variance terms to cancel. When further averaged across $(s, a)$, it is even more likely for this term to be negligible. If the environment is deterministic, then this variance is already zero and there is no approximation.

The use of TD errors for interference is related to previous interference measures based on *gradient alignment*. To see why, notice if we perform an update using one transition $(s_t, a_t, r_t, s'_t)$, then the interference of that update to $(s, a, r, s')$ is $\delta^2(\boldsymbol{\theta}_{t+1}, s, a, r, s') - \delta^2(\boldsymbol{\theta}_t, s, a, r, s')$. Using a Taylor series expansion, we get the following first-order approximation assuming a small step-size $\alpha$:

$$\nabla_{\boldsymbol{\theta}} \delta^2(\boldsymbol{\theta}_t; s, a, r, s')^\top (\boldsymbol{\theta}_{t+1} - \boldsymbol{\theta}_t) = -\alpha \nabla_{\boldsymbol{\theta}} \delta^2(\boldsymbol{\theta}_t; s, a, r, s')^\top \nabla_{\boldsymbol{\theta}} \delta^2(\boldsymbol{\theta}_t; s_t, a_t, r_t, s'_t)$$

This approximation corresponds to *gradient alignment*, which has been used to learn neural networks that are more robust to interference (Lopez-Paz et al., 2017; Riemer et al., 2018). They

measure if this dot product is greater than zero, to determine if there is transfer between two samples; they generally encourage these dot-products to be positive. Other work used gradient cosine similarity, to measure the level of transferability between tasks (Du et al., 2018), and to measure the level of interference between objectives (Schaul et al., 2019). A somewhat similar measure was used to measure generalization in reinforcement learning (Achiam et al., 2019), using the dot product of the gradients of Q functions $\nabla_{\boldsymbol{\theta}} Q_{\boldsymbol{\theta}_t}(s_t, a_t)^\top \nabla_{\boldsymbol{\theta}} Q_{\boldsymbol{\theta}_t}(s, a)$. This measure neglects the direction of the gradients, and so measures both positive generalization as well as interference.

Gradient alignment has a few disadvantages, as compared to using differences in the squared TD errors. First, as described above, it is actually a first order approximation of the difference, introducing further approximation. Second, it is actually more costly to measure, since it requires computing gradients and taking dot products. Computing Update Interference on a buffer of data only requires one forward pass over each transition. Gradient alignment, on the other hand, needs one forward pass and one backward pass for each transition.

## 5 Measuring Catastrophic Interference and Forgetting in RL

In this section, we empirically show that our Interference Across Time measure is correlated with forgetting, where agent performance drops. We define forgetting at each iteration as the difference between the best performance achieved before this iteration, and the performance after the policy improvement step. Precisely, let $\mathbb{E}_{(s,a)\sim d_0}[Q^{\pi_{k+1}}(s, a)]$ be the agent performance after policy improvement step at iteration $k$ where $d_0$ is the start-state distribution, where a random action is taken in the first step. We estimate this value using 100 rollouts. Forgetting from iteration $k$ is defined as

$$\text{Iteration Forgetting}(k) := \max_{i=1,\ldots,k} \mathbb{E}_{(s,a)\sim d_0}[Q^{\pi_i}(s, a)] - \mathbb{E}_{(s,a)\sim d_0}[Q^{\pi_{k+1}}(s, a)].$$

As before, we take the expected tail over all iterations, to measure if the agent suffered from significant forgetting. If even a few iterations involve forgetting, and most steps do not, we should still consider Forgetting to be high. We therefore define Forgetting across iterations as

$$\text{Forgetting} := \mathbb{E}[X | X \geq \text{Percentile}_{0.9}(X)] \text{ for } X = \text{Iteration Forgetting}(K)$$

We chose two environments: CartPole and Acrobot. In CartPole, the agent tries to keep a pole balanced, with a positive reward per step. We chose CartPole because RL agents have been shown to exhibit forgetting in this environment (Goodrich, 2015). In Acrobot, the agent has to swing up from a resting position to reach the goal, receiving negative reward per step until termination. We chose Acrobot because the physical dynamics are similar to Cartpole, but it induces different learning dynamics: instead of starting from a good location, it has to explore to reach the goal.

We ran a variety of agents to induce a variety of different learning behaviors. We generate a set of API and FQI algorithms by varying buffer size $\in \{1000, 5000, 10000\}$, hidden layer size $\in \{64, 128, 256, 512\}$ and number of steps in one iteration $M \in \{100, 200, 400\}$. Each algorithm has 400 iterations. Additionally, we include neural networks with tile coded inputs, which has been shown to break generalization in neural networks (Ghiassian et al., 2020). All experiments are averaged over 30 runs. A buffer for measuring Interference is obtained using reservoir sampling from a larger batch of data, to provide a reasonably diverse set of transitions.

We show the correlation plot between Interference and Forgetting in Figure 1. In all figures except Figure 1 (b), there is a strong correlation between our definitions of Interference and Forgetting. In Figure 1 (b), the correlation is unclear for M=100, possibly because all algorithms do not perform well, with all suffering from similarly high Forgetting which reduces as $M$ increases. We note a few clear outcomes. (1) Neural networks with a smaller hidden size have less interference and forgetting. (2) Large buffer sizes seem to help with small networks. (3) FQI has lower magnitude Interference and less Forgetting than API on both environments. (4) Tile coded inputs reduce Interference and Forgetting for API but not for FQI, possibly because API bootstraps in its targets unlike FQI.

## 6 Mitigating Interference via Online-aware Representations

It is common to assume that the first layers of a neural network play the role of producing a representation, that transforms the inputs into a form that (1) provides suitable abstraction and (2) facilitates

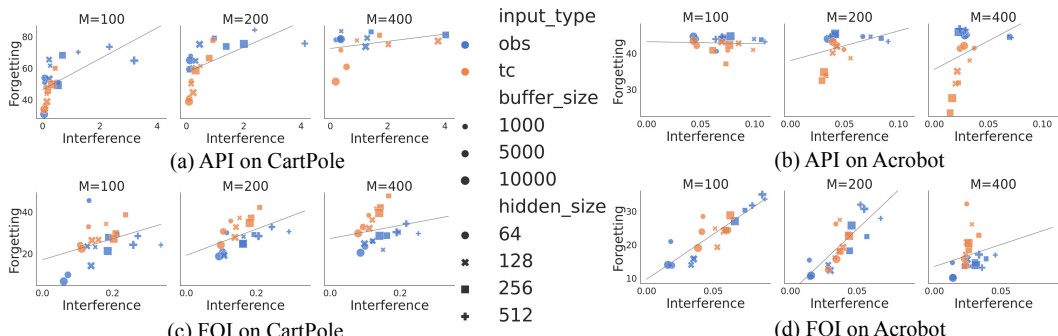

Figure 1: Correlation plot of interference and forgetting. Each point represents one algorithm averaged over 10 runs. Outliers (which have interference larger than 5 and forgetting larger than 80 on CartPole) are excluded from the plots.

learning in the future (typically called transfer). In the online setting, it has also been noted that the representation plays a third role: improving learning under online updating (Sutton, 1996; Liu et al., 2019; Javed and White, 2019). We can view a value function as a two-part approximation with a representation function and a linear weight $Q_{\mathbf{w},\beta}(s,a) := \phi_\beta(s,a)^\top \mathbf{w}$, where $\mathbf{w} \in \mathbb{R}^d$ is the weights in the last layer and $\phi_\beta : \mathcal{S} \times \mathcal{A} \to \mathbb{R}^d$ is the representation learned by the network with weights $\beta$, composed of all the hidden layers in the network. The function $\phi_\beta(s,a)^\top \mathbf{w}$ corresponds to the last layer in the network, with $\mathbf{w}$ the weights of the network.

In this section, we first show that updates in the final layer result in significantly more interference than updates in internal layers. This matches recent insights that learning a sparse representation, offline, can significantly improve stability in online learning (Liu et al., 2019; Javed and White, 2019). Beyond these works, it highlights that the effort spent mitigating interference should be on producing a representation that mitigates interference. We then design an online-aware representation learning approach, that directly minimizes interference.

## 6.1 COMPARING INTERFERENCE DUE TO INTERNAL LAYERS VERSUS THE FINAL LAYER

In this section, we study interference due to updates in the internal layers and the last layer. To study interference separately within the network, we use stochastic block coordinate descent to update $\beta$ and $\mathbf{w}$ separately, as shown in Algorithm 2. We first update $\beta_t$ and compute Update Interference($[\beta_{t+1}, \mathbf{w}_t], [\beta_t, \mathbf{w}_t]$) due to this update. Then we update $\mathbf{w}_t$ and compute Update Interference($[\beta_{t+1}, \mathbf{w}_{t+1}], [\beta_{t+1}, \mathbf{w}_t]$) again. Finally, we compute the Update Interference($[\beta_{t+1}, \mathbf{w}_{t+1}], [\beta_t, \mathbf{w}_t]$) for the whole update, so that we can measure the percentage contribution from each of the separate updates.

We report the percentage contribution of interference in each layer in Figure 2. We found that the last layer exhibits more interference than the internal layers consistently across the size of neural networks. Note that for neural networks with hidden size 512 on CartPole, the internal layers contain about 264,000 parameters while the last layer only has about 2,000 parameters. However, the last layer still exhibits significantly higher interference than the internal layers. This suggests that the internal layers change slowly or they change the TD errors slowly during the optimization process. Examining the distributions of interference over state-action pairs (see Appendix B), we do in fact find that updates in the internal layers result in smaller magnitude interference.

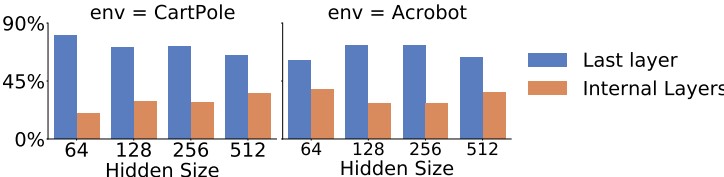

Figure 2: Contribution to Update Interference due to updates of the internal layers versus the last layer. The percentages are averaged over 10 runs.

---

**Algorithm 2** API with Online Representation Learning

---

Initialize weights $\beta, \mathbf{w}$ randomly. Initialize an empty buffer B of size $N$
**for** $t \leftarrow 1, 2, ...N$ **do**
    **If** $t \bmod T_{\text{eval}} = 0$ **then** $Q_k \leftarrow Q_{\boldsymbol{\theta}_t}$, update $\pi_k, b_k$ to use $Q_k$
    Choose $a_t \sim b_k(s_t)$, observe $(s_{t+1}, r_{t+1})$, and add the transition to the buffer
    Sample a mini-batch of transitions $B_t$ from the buffer
    Update $\beta_{t+1}$, using $B_t$ and the representation learning loss (see Appendix A.4)
    $\mathbf{w}_{t+1} = \mathbf{w}_t + \alpha \frac{1}{|B_t|} \sum_{(s,a,r,s') \in B_t} \delta(\beta_{t+1}, \mathbf{w}_t; s, a, r, s') \nabla_{\mathbf{w}_t} Q_{\beta_{t+1}, \mathbf{w}_t}(s, a)$

---

## 6.2 AN ONLINE-AWARE REPRESENTATION LEARNING ALGORITHM

In this section, we develop an algorithm to explicitly learn representations that mitigate interference. The idea is simple, and actually allows a variety of offline representation learning approach to be incorporated into API and FQI. The first step is to separate the update to the representation, using a block coordinate update, as we used in the last section to measure interference in different layers. The second is to incorporate a representation learning loss, which can be update as it would be offline simply by using a replay step from the replay buffer. This allows us to both incorporate two recently proposed offline objectives: Sparse Representation NN (SRNN) which uses a distributional regularizer on the hidden nodes (Liu et al., 2019) and Online-Aware Meta-Learning (OML) which uses a meta-learning approach to adjust the representation to minimizes the loss after $n$ online updates (Javed and White, 2019). To the best of our knowledge, this is the first time these two approaches have been extended to the online setting, which the representation being learned concurrently with the value estimates.[1] We summarize this generic approach in Algorithm 2, with more specific details for each representation learning approach in Appendix A.4 and A.5.

We can adapt the objective from Javed and White (2019) to minimize Update Interference in RL. The goal is to learn a representation to minimize Update Interference from $n$ updates. We can sample a random evaluation buffer $B$, from the larger replay buffer, that implicitly specifies the distribution $d$ in the Update Interference. Then we get $\mathbf{w}_{t+n}$ the weights after $n$ experience replay updates, for a fixed representation $\beta_t$. We get the resulting Update Interference$([\beta_t, \mathbf{w}_{t+n}], [\beta_t, \mathbf{w}_t] = \sum_{i=1}^{|B|} \delta_i([\beta_t, \mathbf{w}_{t+n}])^2 - \delta_i([\beta_t, \mathbf{w}_t])^2$. The goal is to optimize the current representation $\beta_t$ so it produces the lowest interference after these $n$ updates to $\mathbf{w}_t$. Because we do not update $\mathbf{w}_t$ from $\beta_t$ in the second term, we only need to adjust $\beta_t$ for the first term. Effectively, we are adjusting $\beta_t$ so that it allows for a better final $\mathbf{w}_{t+n}$. This is precisely how OML was defined, but for a supervised loss. In this sense, in its standard form, it is already implicitly minimizing interference.

## 6.3 EMPIRICAL COMPARISON

We compare FQI, API-ORL, API-SRNN and API-OML on three classic control problems. API-ORL simply involves learning $\beta_t$ with the two stage procedure described in Section 5.1, without special representation learning approaches. Figure 3 summarizes the performance in Acrobot. The improvement

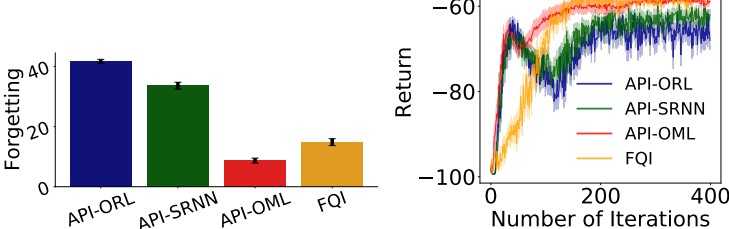

(a) Total Forgetting across iterations.    (b) Expected return vs iterations.

Figure 3: Acrobot results, averaged over 30 runs with standard errors.

for API-OML is made even more apparent when looking at Forgetting—which reflects stability. The results for Mountain Car and Cartpole can be found in Appendix C. All results are averaged over 30 independent runs and learning curves and bar plots show standard error.

---

[1]A related approach, Meta-Experience Replay (MER) (Riemer et al., 2018), directly optimizes gradient alignment. MER does not explicitly learn a representation. Instead, it updates the entire network in the inner loop and outer loop. As a result, it is significantly more computationally intensive. This necessitates the use of a first-order approximation, and we were unable to get it to perform consistently.

Figure 3b suggests our new API-OML method is better than the baselines, but averaging over runs actually removes much of the interesting structure, which is particularly relevant when investigating interference. Looking closer, Figure 4a shows the return per run, revealing that API-ORL has considerable problems learning and maintaining stable performance. API-OML in comparison is substantially more stable and reaches higher performance. Figure 4b shows the iteration interference. Overall API-OML exhibits far less interference, but does exhibit high interference in two runs. These runs exactly correspond to the two runs in Figure 4a where API-OML is less stable in the beginning of learning. We do not know exactly why OML struggles in 2 of the 30 runs, but our measure of interference allows us to pinpoint precisely when it occurs. Finally, in Figure 3a we see that API-OML exhibits significantly less forgetting than all three baselines, dramatically improving over API-ORL.

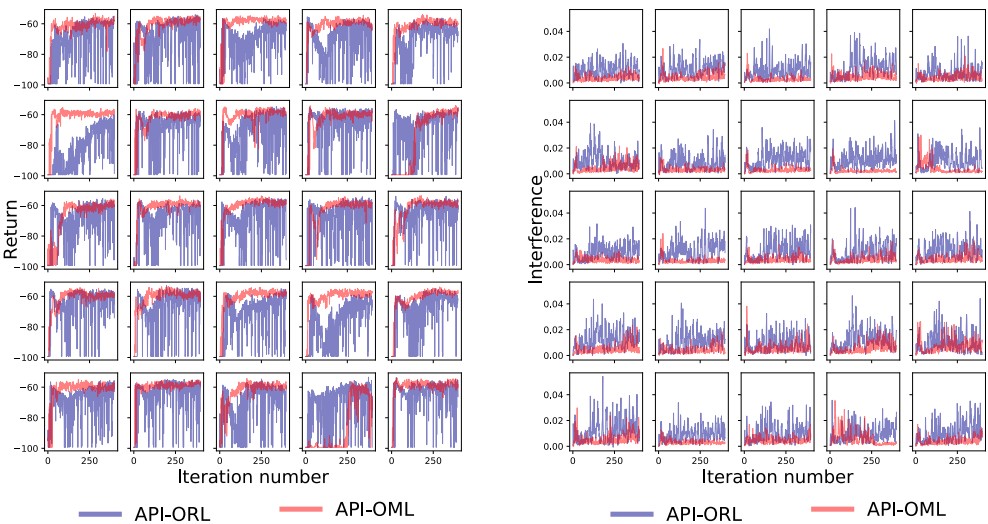

(a) Learning curves for API-ORL and API-OML, one run per subplot.

(b) Iteration Interference for API-ORL and API-OML.

Figure 4: Learning curves and Iteration Interference for API-ORL and API-OML in Acrobot.

## 7  CONCLUSION

In this paper, we proposed a definition of interference for iterative value-based methods, API and FQI, and justified the use of squared TD error to approximate this interference. We showed this interference measure is correlated with control performance. We additionally provided some insights into interference in deep reinforcement learning algorithms, namely that FQI has lower interference than API and that API—which uses bootstrapping rather than the fixed targets in FQI—benefits more from strategies that reduce generalization in inputs. We then used the measure to show that the interference in updates, for standard neural networks, is due more to updates in the final layer than due to updates to internal layers. This motivated extending API and FQI to include an a strategy to do online-aware representation learning, where the representation is explicitly trained to mitigate interference. The new algorithm provides a simple mechanism to convert two existing offline representation learning approaches to the online setting. We concluded with a demonstration that the resulting algorithm, called API-OML, significantly improved stability, and exhibited low interference under our measure as well as low forgetting.

These results highlight several promising avenues for improving stability in RL. One potentially surprising outcome was the instability, within a run, of a standard method like API. The learning curve for API actually looked reasonable, and without examining individual runs, this instability would not be obvious. This motivates re-examining many of our agents with alternative measures, like forgetting and other measures of stability. It also highlights that there is an exciting opportunity to significantly improve our agents, by focusing on these robustness improvements.

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

# A EXPERIMENTAL DETAILS

## A.1 EXPERIMENT SETUP

We experiment with three environments: CartPole, Acrobot and Mountain Car from the OpenAI gym (https://gym.openai.com/). We set the maximum steps per episode to 500, and the number of training iterations to 400. We use a discounting factor $\gamma = 0.99$ in all environments. CartPole, Acrobot, and Mountain Car have 4, 6, and 2 dimensional states respectively. For tile coded inputs, we use 4 tiles and 8 tilings for each dimension of the state seperately, which results in input size of 160 and 240 respectively in CartPole and Acrobot.

We use 40 Monte Carlo rollouts to estimate the performance of the policy at each iteration $\mathbb{E}_{(s,a)\sim d_0}[Q^\pi(s,a)]$. For evaluating the TD error difference, we use a reservoir buffer of size 1000, which approximates uniform sampling from all the past transition. For the experiment in Section 5, Interference Across Iterations and Forgetting are computed over the last 200 iterations.

## A.2 NETWORK ARCHITECTURE AND HYPERPARAMETERS

For all experiments, we use a two-layer neural network with ReLU activation, He intialization to initialize the neural networks, and Adam optimizer.

For the experiments in Section 4, we generate a set of hyper-parameter $\Theta$ by choosing each parameter in the set:

- batch size $= 64$
- Step size $\alpha = 0.0003$
- buffer size $\in \{100, 5000, 10000\}$
- Hidden size $\in \{64, 128, 256, 512\}$
- Number of step in an iteration $M \in \{100, 200, 400\}$
- Numver of iteration $= 400$
- Input type $\in \{\text{state}, \text{tile coded state}\}$

For the experiments in Section 5, we sweep the hyperparameter in the range:

- Batch size $= 64$
- Step size $\alpha \in \{0.001, 0.0003, 0.0001\}$
- Hidden size $= 128$
- Buffer size $\in \{1000, 5000, 1000\}$

The parameters are chosen based on average performance over the last 200 iterations.

## A.3 ITERATIVE VALUE ESTIMATION ALGORITHMS WITH SEPARATE UPDATES

We provide a complete description of the algorithm in Algorithm 3.

## A.4 ONLINE-AWARE META-LEARNING

This OML objective can be optimized similarly to other gradient-based meta-learning objectives (Finn et al., 2017), and can be implemented in API and FQI. We provide a complete description of the algorithm in Algorithm 4. Note that the original algorithm from Javed and White (2019) samples $B_1, \ldots, B_I$ in a sequential manner, that is, $B_1, \ldots, B_I = (s_{t+1}, a_{t_1}, r_{t+1}, s'_{t+1}), \ldots, (s_{t+I}, a_{t_I}, r_{t+I}, s'_{t+I})$ is a trajectory from the buffer, but we sample each of $B_1, \ldots, B_I$ as a random mini-batch of transitions from the replay buffer.

In our experiment, We sweep over the hyperparameters in the set:

- Inner update optimizer $=$ SGD

- Meta update optimizer = Optimizer for the last layer = Adam
- $\alpha \in \{0.001, 0.0003, 0.0001\}$
- $\alpha_{\text{inner}} \in \{0.01, 0.001, 0.0001, 0.00001\}$
- Number of inner updates $K \in \{10, 20, 40\}$

### A.5 SPARSE REPRESENTATION NEURAL NETWORKS

Liu et al. (2019) use the distributional regularizers to learn sparse representation neural networks (*SRNN*). If the vanilla neural network optimizes the objective $J$, SRNN simply adds a regularization:

$$J_{SRNN}(\beta, \mathbf{w}) = J(\beta, \mathbf{w}) + \lambda_{SKL} \sum_{j=1}^{d} SKL_\theta(\bar{\phi}_{\beta,j})$$

where $SKL(\bar{\phi})$ is a component-wise regularization on the expected activation $\bar{\phi} = \sum_{(s,a)\sim B} \phi(s,a)$ and $\phi_j$ denote the $j$-th component of $\phi$. $\lambda_{SKL}$ controls the weight on the regularization and $\theta$ control the sparsity level. SRNN can be implemented in API and FQI by simply adding a regularization when updating the parameter $\beta$.

In our experiment, we sweep over the key hyper-parameters of SRNN in the set:

- Adam optimization
- $\alpha \in \{0.001, 0.0003, 0.0001\}$
- $\lambda_{SKL} \in \{0.1, 0.01, 0.001, 0.0001\}$
- $\theta \in \{0.1, 0.2, 0.4\}$

## B  DISTRIBUTION OF ACCURACY CHANGE

In this experiment, we visualize the distribution of Accuracy Change over state-action pairs. We run the API-Stage algorithm on CartPole for 40k steps and choose three time steps corresponding to the time step (1) when the average Accuracy Change is the largest (2) when the average Accuracy Change is the smallest (3) when average Accuracy Change is the closest to 0.

We show the distribution of Accuracy Change due to the internal layers and the last layer, for the three time step we choose, in Figure 5. The results suggests that updates in the internal layers has smaller magnitude of interference than updates in the last layer for all the three cases.

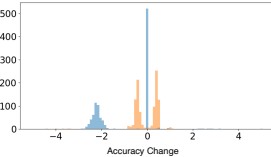 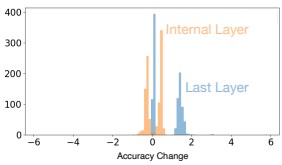 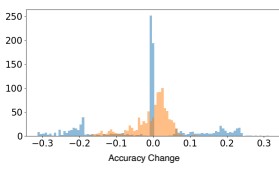

(a) Average is highly positive.   (b) Average is highly negative.   (c) Average is close to zero.

Figure 5: Histogram of Accuracy Change in CartPole.

## C  ADDITIONAL EXPERIMENT RESULTS OF SECTION 6

We compare the algorithms mentioned in Section 6 in CartPole, Acrobot and Mountain Car, and show the learning curve in Figure 6. We found that OML significantly improve stability and efficiency compared to API-ORL and API-SRNN in Acrobot and Mountain Car. All algorithms perform similarly in CartPole, possibly because they all choose the smallest step size in the range in this domain.

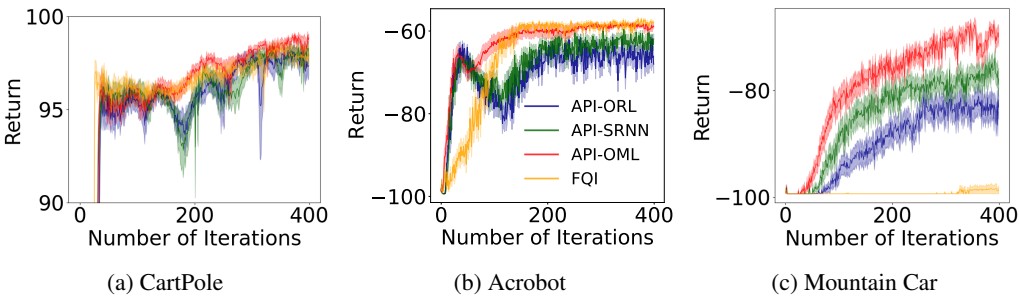

(a) CartPole  (b) Acrobot  (c) Mountain Car

Figure 6: Learning curve of API with different representation learning algorithms and FQI. The number are averaged over 30 runs with one standard error.

---

**Algorithm 3** Iterative Value Estimation with separate updates
---

Initialize weights $\boldsymbol{\theta}_0$. Initialize an empty buffer of size $B$.
**for** $t \leftarrow 1, 2, ...N$ **do**
  **If** $t \bmod T_{\text{eval}} = 0$ **then** $Q_k \leftarrow Q_{\boldsymbol{\theta}_t}$, update $\pi_k$ to be greedy w.r.t $Q_k$, $b_k$ to be $\epsilon$-greedy
  Choose $a_t \sim b_k(s_t)$, observe $(s_{t+1}, r_{t+1})$, and add the transition to the buffer
  Sample a mini-batch of transitions $B_t$ from the buffer and update the the internal layers
  $\beta_{t+1} = \beta_t + \alpha \frac{1}{|B_t|} \sum_{(s,a,r,s') \in B_t} \delta(\beta_t, \mathbf{w}_t; s, a, r, s') \nabla_{\beta_t} Q_{\beta_t, \mathbf{w}_t}(s, a)$
  Update the the last layer
  $\mathbf{w}_{t+1} = \mathbf{w}_t - \alpha \frac{1}{|B_t|} \sum_{(s,a,r,s') \in B_t} \delta(\beta_{t+1}, \mathbf{w}_t; s, a, r, s') \nabla_{\mathbf{w}_t} Q_{\beta_{t+1}, \mathbf{w}_t}(s, a)$

---

---

**Algorithm 4** Iterative Value Estimation with online-aware representation learning
---

Initialize weights $\boldsymbol{\theta}_0$. Initialize an empty buffer of size $B$.
**for** $t \leftarrow 1, 2, ...$ **do**
  **If** $t \bmod T_{\text{eval}} = 0$ **then** $Q_k \leftarrow Q_{\boldsymbol{\theta}_t}$, update $\pi_k$ to be greedy w.r.t $Q_k$, $b_k$ to be $\epsilon$-greedy
  Choose $a_t \sim b_k(s_t)$, observe $(s_{t+1}, r_{t+1})$, and add the transition to the buffer
  *// Representation inner update:*
  $\mathbf{w}_{t,0} \leftarrow \mathbf{w}_t$
  **for** $i \leftarrow 1, 2, ...I$ **do**
    Sample transitions a mini-batch $B_i$ from the buffer
    $\mathbf{w}_{t,i} \leftarrow \mathbf{w}_{t,i-1} - \alpha_{\text{inner}} \frac{1}{|B_i|} \sum_{(s,a,r,s') \in B_i} \delta(\beta_t, \mathbf{w}_{t,i-1}; s, a, r, s') \nabla_{\mathbf{w}_{t,i-1}} Q_{\beta_t, \mathbf{w}_{t,k-1}}(s, a)$
  Sample a mini-batch of $B_t$ from the buffer
  *// Representation meta update:*
  $\beta_{t+1} = \beta_t + \alpha \frac{1}{|B_t|} \sum_{(s,a,r,s') \in B_t} \delta(\beta_t, \mathbf{w}_{t,I}; s, a, r, s') \nabla_{\beta_t} Q_{\beta_t, \mathbf{w}_{t,I}}(s, a)$
  *// Last layer update:*
  $\mathbf{w}_{t+1} = \mathbf{w}_t - \alpha \frac{1}{|B_t|} \sum_{(s,a,r,s') \in B_t} \delta(\beta_{t+1}, \mathbf{w}_t; s, a, r, s') \nabla_{\mathbf{w}_t} Q_{\beta_{t+1}, \mathbf{w}_t}(s, a)$

---

