# OpenReview forum: "Measuring and mitigating interference in reinforcement learning"
_ICLR.cc/2021/Conference — Reject_

### Official Review · AnonReviewer4 · 2020-10-26
**Interesting work, with weak theoretical motivation**

**Rating:** 5
**Confidence:** 4

**Review:**

This paper studies the reason for interference, aka catastrophic forgetting, when using parametric models for Reinforcement Learning. The authors draw the connection with previous methods and introduce some reasonable measure of interference. Then, they introduce a method to explicitly address the problem of interference, showing some good empirical results w.r.t. selected baselines.

I think that this paper studies an interesting problem, but its analysis is a bit superficial and not supported by rigorous theoretical analysis. The proposed measures of interference make sense, but it seems to me that they are not reliable measures, especially in the considered online learning setting. In fact, the TD-error may increase significantly across several iterations because of the agent exploring unvisited states, and not only because of interference. Previous works, e.g. Prioritized Experience Replay (Schaul et al, 2016), show that a higher TD-error is actually desirable to guide exploration, so I'm unsure how the motivation behind this work relates with the literature. Moreover, since catastrophic forgetting is mostly problematic in deep RL, I'd have liked a stronger focus on deep RL, where the author could have tested the benefit of their approach on several algorithms based on the TD error, e.g. DQN, DDPG, TD3, SAC. It is also confusing how the authors refer to FQI as an online algorithm. FQI is known to be a batch RL algorithm, i.e. an offline algorithm where a fixed dataset of transitions collected by another agent is available, even though it can be used for iterative policy updates, but this is not the typical scenario of FQI.

Without an extensive empirical analysis, and considering the absence of a rigorous theoretical analysis motivating the proposed interference measure, I think this paper is not ready for publication and I encourage the authors to improve it, especially showing stronger and more significant empirical evidence over representative baselines, perhaps even considering some multi-task and/or lifelong learning problems where interference constitutes a bigger issue.

Pros
------
* The considered problem is interesting for a broad group of researchers in RL;
* The presentation is clear enough.

Cons
-------
* The proposed measures are intuitive, but not supported by strong theoretical guarantees. In particular, the effect of exploration on the behavior of the TD error is not accurately discussed;
* Absence of deep RL experiments to show the effectiveness of the proposed approach on more challenging problems.


Post-rebuttal feedback
-------------------------------
I thank the authors for their reply. I still think that this paper has the major problem of presenting results about interference that are not strong enough to be published. In particular, I agree when the author say that "the paper tries to bring conceptual clarity to this important topic, and provide a clear empirical methodology to measure interference and understand correlations to forgetting", but still I think that the overall contribution consists "just" of one (of several other possible ones) intuitive method to measure interference, without a solid motivation. To me, this paper is promising but should present more significant results to have a stronger impact. I think the options are only two: stronger theoretical motivation, larger empirical analysis especially considering deep RL. Between the two, I consider the second option the best.

---

> ### Author Response · Authors · 2020-11-19
> **Response to R4 (1/2)**
>
> Thank you for the comments.
>
> > “I think that this paper studies an interesting problem, but its analysis is a bit superficial and not supported by rigorous theoretical analysis.”
>
> The RL community has yet to converge on a clear definition of interference. For example a recent ICML paper proposes several different ways to measure interference (https://arxiv.org/pdf/2003.06350.pdf). We provide a formalization and definition that can be measured, correlated with other performance measures and optimized (in addition to reward). No previous work has done these things.
>
> This is a relatively unexplored topic in RL. It confuses us why one paper should be on the hook to provide a definition, demonstrate it makes sense, show it helps as part of an objective---and demonstrate the new algorithm at scale and provide a new theoretical framework.
>
> Theory linking interference to control performance is absolutely of interest. However, we as yet do not even have clear definitions or empirical work to provide clarity on the role of interference. It is hard to even know where to start theoretically, what questions to answer. We believe a purely empirical work, formalizing how to measure interference and examining its role, is an important first step.
>
> > “The TD-error may increase significantly across several iterations because of the agent exploring unvisited states, and not only because of interference.”
>
> First we would like to point out (as we detailed in the introduction) that we study interference in the policy evaluation step of policy improvement---not in the usual online RL setting of prior work. We explicitly choose to investigate interference in the GPI setting where the agent alternates between estimating the value function from data (policy evaluation) and greedifiing the policy and collecting a new batch of data. This avoids the complexities of the policy and value function changing on each step, and the confounding factor of exploration.
>
> For policy evaluation, the average TD-error across state-actions does not have to increase irrespective of the exploratory behavior of the policy. This is the primary motivation behind defining interference in the policy evaluation setting.
>
> > “Prioritized Experience Replay (Schaul et al, 2016), show that a higher TD-error is actually desirable to guide exploration.”
>
> It’s unclear what this means. PER, primarily, makes learning more sample efficient by focusing on transitions that have high TD error. The PER paper does talk about the potential of using signals from PER for guiding exploration in section 6 (the argument being that if experience generated from exploration strategy x is found to be more useful than that generated by y, x is a better exploration strategy), but they do not empirically verify if their proposal is useful and as a result, never show that “a higher TD-error is actually desirable to guide exploration.”
>
> > “Moreover, since catastrophic forgetting is mostly problematic in deep RL, I'd have liked a stronger focus on deep RL, where the author could have tested the benefit of their approach on several algorithms based on the TD error, e.g. DQN, DDPG, TD3, SAC.”
>
> Catastrophic forgetting has been clearly demonstrated in one-hidden layer neural networks. In fact, the term catastrophic forgetting predates deep-RL or even deep learning (see literature by R.M French or McCloskey). Moreover, DQN, TD3, and SAC introduce additional challenges for measuring interference. It is better to investigate a problem in the simplest setting it arises in. Our first step in the paper is to study interference in the GPI setting. Other methods, including DDPG, TD3 and SAC, are beyond the scope of the paper and they are a next step to extend beyond our setting.

---

> > ### Author Response · Authors · 2020-11-19
> > **Response to R4 (2/2)**
> >
> > > “I encourage the authors to improve it, especially showing stronger and more significant empirical evidence over representative baselines.”
> >
> > The dichotomy given here is that we should either have a rigorous theory paper or a large empirical paper demonstrating the utility of the algorithm. Instead, we have a third option. The paper tries to bring conceptual clarity to this important topic, and provide a clear empirical methodology to measure interference and understand correlations to forgetting. This first small empirical study itself is primarily a case study, but also has some insights about the role of buffer size, input type and hidden layer size on interference and forgetting. We then use the conceptual clarity on interference to design a new algorithm, directly mitigating interference, and demonstrate that such an approach is promising for stabilizing algorithms. Here again an important (and maybe surprising) result is just how unstable API is, and how much use a learned representation to mitigate interference increases stability (see Figure 4).
> >
> > Further, when it comes to representative baselines, we make the same comment as to Reviewer 1. In the second part of the paper, we study interference for representation learning algorithms. Note that our goal is to investigate if mitigating interference can improve performance, rather than to compare more generally to algorithms designed to reduce forgetting. To the best of our knowledge, we do not know of any online representation learning methods that have been shown to reduce interference. Most representation learning approaches have used offline representation learning, including SRNN and OML. Our framework is a generic approach to incorporate these offline approaches, into an online setting.

---

### Official Review · AnonReviewer3 · 2020-10-27
**An interesting paper studying the interference under an API and FPI setting, but more clarification required.**

**Rating:** 6
**Confidence:** 2

**Review:**

This paper studies the the interference problem under the API and FPI setting. It designs new measure for interference, and shows that the interference measure is correlated with forgetting. With the help of the interference measure, the paper studies the importance of the final layer of the neural network and proposes a new algorithm to mitigate interference.

Overall, the paper is well-written but I have some questions regarding the logic of the paper. First, it's not super clear to me what is the role of the forgetting measure. Since the validation of interference measure is evaluated by forgetting, I'm assuming forgetting is a more commonly used measure (please correct me if I'm wrong).
- If this is the case, then it seems the importance of interference measure is to come up with the SRNN and OML variants. But I don't get why we cannot use the forgetting measure for regularization, maybe some clarifications needed here. If the interference measure is not necessary for OML and SRNN, I don't quite get the necessarily of the interference measure.
- If this is not the case, then why forgetting is an interesting measure to compare with?

In addition, I think it may be easier to follow if the unused definitions can be moved to Appendix, e.g., Pointwise Interference and Interference Across Iterations are only mentioned when they're defined, On the other hand, I think it'll be helpful to move some contents from Appendix to Sec 6.2, since it is the place that explicitly take advantage of the interference measure.

Lastly, there two minor points:
- Line 4. the definition of iid is missing
- Algorithm 1: it seems the the definition of b_k is missing

---

> ### Author Response · Authors · 2020-11-19
> **Response to R3**
>
> Thank you for the comments.
>
> > “I'm assuming forgetting is a more commonly used measure (please correct me if I'm wrong).”
>
> The intuitive definition of forgetting is the difference between the best performance achieved before time t and performance at time t. Some variants of the intuitive definition are used in the multi-task supervised learning community (e.g., [1], [2]). In RL, forgetting can be defined in terms of the agent’s control performance, instead of prediction accuracy, and that is the forgetting measure we used in the paper. We will explain these connections, and add these citations, in the paper.
>
> [1] Chaudhry, Arslan, Puneet K. Dokania, Thalaiyasingam Ajanthan, and Philip HS Torr. "Riemannian walk for incremental learning: Understanding forgetting and intransigence."
>
> [2] Serra, Joan, Didac Suris, Marius Miron, and Alexandros Karatzoglou. "Overcoming catastrophic forgetting with hard attention to the task."
>
> > “But I don't get why we cannot use the forgetting measure for regularization, maybe some clarifications needed here.”
>
> Not all measurable quantities are reasonable to optimize. We introduce forgetting because it is of interest to measure, but it makes little sense to optimize. Optimizing forgetting is relative to the agent’s own performance in the past, which is somewhat circular. But in any case, it is optimized by simply trying to find the optimal policy, which is at least as good as any policy in the past. Optimizing forgetting would in fact tell the agent nothing about interference or generalization, and our goal is to obtain an objective to reduce interference and increase positive generalization.

---

### Official Review · AnonReviewer2 · 2020-10-28
**Probably interesting idea but not convincing enough**

**Rating:** 4
**Confidence:** 3

**Review:**

This paper proposes new measures to quantify interference in reinforcement learning at different granularity and show the correlation between interference and forgetting, and suggests to use extra representation loss to reduce interference.

In general, the definition of  interference at different granularity looks reasonable and the relation with forgetting matches intuition. However, there are no experiments to show any difference between those interference, such as update interference,  iteration interference, and interference across iterations.  So what's point to define interference with different granularity if only one is used in the end?

The authors propose using extra representation loss to reduce interference, but the motivation is rather vague.  As the experiments show that internal layers contribute much less to interference, why it is better to adjust internal layers by representation loss than adjust the last layer which has much more influence on interference?  Is there more solid justification to connect representation loss and interference?

The authors also introduces three different types of representation loss and OML obviously outperforms the others, but there is no enough analysis about why OML is supreme and then no guidance of how to choose a proper representation loss.

As I'm not an expert in reinforcement learning, I will leave the novelty and significance to other reviewers to decide.

Some minor issues:
1. Equations should be numbered.
2. Could authors elaborate a bit on how to get the last equation on page 4? As the last term in the right side hasn't appeared before.
3. The coefficients of forgetting and interference in the subfigures of Fig. 1 should be provided.

---

> ### Author Response · Authors · 2020-11-19
> **Response to R2**
>
> > “So what's point to define interference with different granularity if only one is used in the end?”
>
> In order to define Interference Across Iterations, we need to define Iteration Interference, to summarize interference over time, and Update Interference, to summarize interference over state-action pairs. It conceptually leads to the one we measure. It is still worthwhile highlighting these differences, even if we don't measure at that granularity. When others use these definitions, they in fact could measure at that granularity.
>
> > “As the experiments show that internal layers contribute much less to interference, why it is better to adjust internal layers by representation loss than adjust the last layer which has much more influence on interference?”
>
> The interference in the last layer can be attributed to the features used by the last layer, and the weights of the last layer. The meta-objective, as a result, can be minimized by (1) updating the features, (2) updating the weights of the last layer, or (3) updating both the features and the weights. We chose to only optimize the features by minimizing interference because earlier work has shown that interference in the last layer can be greatly reduced by just learning the right features. This was demonstrated by SRNN [1], OML [2] and ANML [3] that showed that for the right feature representations, the last layer can learn complicated tasks with essentially no interference.
>
> Additionally, minimizing interference is not the only goal of the agent. The actual goal of the agent is to learn the optimal value function while minimizing interference. This means that two different objectives --- one minimizing the td error and the other minimizing the increase in td-error  --- must be optimized simultaneously.
>
> Now in principle, we could have optimized the weights of the last layer to minimize interference as well, as suggested by the reviewer. But given the strongs results of [1], [2] and [3] and the fact that the last layer is learning the value function --- the primary goal of the agent ---  we concluded that it is better to update only the features using the meta-objective.
>
> [1] Liu, Vincent, et al. "The utility of sparse representations for control in reinforcement learning." Proceedings of the AAAI Conference on Artificial Intelligence. 2019.
>
> [2] Javed, Khurram, and Martha White. "Meta-learning representations for continual learning." Advances in Neural Information Processing Systems. 2019.
>
> [3] Beaulieu, Shawn, et al. "Learning to continually learn." arXiv preprint arXiv:2002.09571. 2020.
>
> > The authors also introduce three different types of representation loss and OML obviously outperforms the others, but there is no enough analysis about why OML is supreme.
>
> In section 6.2, we show that OML implicitly minimizes the interference measure but the other two representations do not. We believe that explains the stable performance of OML.

---

### Official Review · AnonReviewer1 · 2020-10-29
**An important problem but the contributions have limited novelty, no comparison to the related work**

**Rating:** 5
**Confidence:** 4

**Review:**

Summary

The paper studies interference and forgetting in the context of reinforcement learning (RL).  On the example of the Iterative Value Estimation family of algorithms, the authors define interference as the increase of the true Q target prediction error after updating Q function parameters. Since the true Q target is usually unknown, the authors propose to use the difference of squared TD-errors between updates as a proxy for interference. The paper further defines forgetting as the difference between the current agent performance and the best performance across all previous updates. On CartPole and Acrobot environments, the authors show a positive correlation between the proposed measures of interference and forgetting. They further qualitatively demonstrate that updates of the last layer weights result in higher interference compared to updates of intermediate layers. Finally, the authors propose an algorithm based on meta-learning for learning representations that minimize interference resulting in more stable return plots on Acrobot.

Strengths
- The topic brought in the paper is important: the distribution shift and moving targets in RL are indeed problematic and often lead to instabilities during training and forgetting of high-return policies.
- The paper is written clearly and generally easy to follow.

Weaknesses
- The proposed algorithm is designed to explicitly minimize interference. However, seeking for minimizing interference / forgetting on its own might prohibit exploration. For example, an agent that achieves the worst possible returns and is not learning at all will have zero interference / forgetting. (Perhaps it is more reasonable to seek monotonic policy improvement?)
- The finding that changes in parameters of the last layer result in higher interference seems unsurprising as, generally, changes in the final layer parameters affect the output of a neural network more than changes in intermediate layers.
- The proposed method for learning representations is based on meta-learning. It is unclear whether the learning curves on Acrobot are more stable due to claimed minimization of interference or due to using a more powerful optimization method. An ablation study will improve the quality of the paper.
- The authors should consider using harder environments to make the experimental results more convincing.

Positioning relative to the literature
- One of the main methodological contributions of the paper is using squared TD-error as a proxy for measuring interference. At the end of section 4, the authors mention related methods based on first-order approximation of interference. However, no comparison with the methods was provided.
- Similarly, the authors do not compare with other methods that address the forgetting in the RL context. For example, [2] uses simple weight averaging for minimizing forgetting and achieves learning curves similar to the ones in Figure 4.
- Perhaps the authors should reconsider the term “interference”. The cited paper [1] uses the term “interference” for an inner product between gradient estimates evaluated at different data points and seeks, in contrast, to maximize interference.

Recommendation

The reviewer leans towards rejecting the paper. The discussed problem is important, however, the findings of the paper do not seem generally surprising. The authors claim that “there is no established online measure of interference for RL” but the resulting measure of interference is simply a difference between squared TD-errors before and after an update. The proposed measure of forgetting, being a difference of current returns and previous best returns, has limited novelty too. Moreover, the paper lacks a comparison with related work. Addressing the outlined weaknesses might increase the assigned score.

**

Post-rebuttal update: the score is increased from 4 to 5. See the response to the authors' comments.

**

[1] Bengio, Emmanuel, Joelle Pineau, and Doina Precup. "Interference and Generalization in Temporal Difference Learning." arXiv preprint arXiv:2003.06350 (2020).

[2] Nikishin, Evgenii, Pavel Izmailov, Ben Athiwaratkun, Dmitrii Podoprikhin, Timur Garipov, Pavel Shvechikov, Dmitry Vetrov, and Andrew Gordon Wilson. "Improving stability in deep reinforcement learning with weight averaging." In Uncertainty in artificial intelligence workshop on uncertainty in Deep learning. 2018.

---

> ### Author Response · Authors · 2020-11-19
> **Response to R1 (1/2)**
>
> Thank you for the comments.
>
> > “The proposed algorithm is designed to explicitly minimize interference. However, seeking for minimizing interference / forgetting on its own might prohibit exploration. For example, an agent that achieves the worst possible returns and is not learning at all will have zero interference / forgetting.”
>
> We agree that minimizing interference alone is not sufficient for learning, and the real goal is to minimize interference while maximizing generalization. Our proposed algorithm is doing exactly that. First, our algorithm is updating the value estimates using API or FQI using a non-zero step-size at every-step. This forces the system to minimize interference while also learning. Secondly, API-OML minimizes Update Interference metric that is not clipped to be non-negative. This allows our meta-objective to go beyond making interference zero (because it can make it negative) and avoid the pitfall of minimizing interference by turning off learning.
>
> We concede that this was unclear from the write-up, as we defined Update Interference metric to be non-negative but then used the unclipped version. We will fix this issue.
>
> That being said, we want to clarify that minimizing interference would not prohibit exploration in our setting. This is because within an iteration, the policy is fixed (Policy evaluation). Even if the policy is highly exploratory, the average td error across the state-space could go down for the right representation.  Exploration is, in-fact, tangential to interference for policy evaluation.
>
> > “The finding that changes in parameters of the last layer result in higher interference seems unsurprising as, generally, changes in the final layer parameters affect the output of a neural network more than changes in intermediate layers.”
>
> Even if the findings are unsurprising, they have not been shown by prior work, to the best of our knowledge. Verifying untested ideas is an important aspect of research and science even if the said ideas appear intuitively true. If you can point to prior work that has a clear empirical demonstration of this phenomenon in RL we are excited to discuss it with you.
>
> > “It is unclear whether the learning curves on Acrobot are more stable due to claimed minimization of interference or due to using a more powerful optimization method. An ablation study will improve the quality of the paper.”
>
> We are not sure what the reviewer meant by "more powerful optimization method." The meta-objective is being optimized using SGD, similar to the baselines.
>
> Perhaps the reviewer meant that the meta-learning algorithm requires more compute and memory and is therefore a more powerful method? If so, then while we agree that the API-OML requires more memory and compute, that is not necessarily bad. An algorithm that can benefit from more resources and learn better is a more scalable and useful algorithm. One possible option would be to run API for more iterations, to benefit from more compute and see if this closes the gap. However, we can already see in later learning that API remains much less stable than API-OML.
>
>
> > “The authors should consider using harder environments to make the experimental results more convincing.”
>
> We designed our experiments to study interference in the simplest cases it arises. This is a common approach, see Sutton’s black and white world used to study tracking [1] and b-suite [2]. They used small environments because they were the right choice for their scientific questions. Larger and more complex domains almost certainly introduce confounding factors that are both unknown and difficult to control. The environments we used cause failures in API/FQI, making them a suitable place to investigate these measures of interference. Scaling up is one scientific question, but not the only one and not always the first one to ask.
>
> [1] Sutton, Richard S., Anna Koop, and David Silver. "On the role of tracking in stationary environments."
> [2] Osband, Ian, Yotam Doron, Matteo Hessel, et al. "Behaviour suite for reinforcement learning."
>
> > “The authors mention related methods based on first-order approximation of interference. However, no comparison with the methods was provided.”
>
> We mention the disadvantage of gradient alignment in the last paragraph of Section 4. It is actually a first-order approximation of the difference and more costly to compute. There is already an efficient approximation of interference, so we think it is unnecessary to compare to expensive methods introducing further approximation.

---

> > ### Author Response · Authors · 2020-11-19
> > **Response to R1 (2/2)**
> >
> > > “Similarly, the authors do not compare with other methods that address the forgetting in the RL context.”
> >
> > Our goal is to investigate if mitigating interference using representation learning can improve performance and reduce forgetting, rather than to compare more generally to algorithms designed to reduce forgetting. To the best of our knowledge, we do not know of any online representation learning methods that have been shown to reduce interference. Most representation learning approaches have used offline representation learning, including SRNN and OML. Our framework is a generic approach to incorporate these offline approaches, into an online setting.
> >
> > The cited work from Nishkin et al. is not attempting to mitigate interference. Rather, it tries to mitigate forgetting similar by using weight averaging. There are several strategies that can be taken to reduce forgetting, including learning many neural networks and freezing them; lots and lots of replay with a very large buffer; identifying parameters important for performance and regularizing them from changing (Elastic weight consolidation, Synaptic intelligence); learning multiple networks and selecting the best one for making predictions and learning (Mixture of experts); and learning a parametric model of the data distribution (e.g., GAN) and using it to generate data for replay. Likely a final solution to forgetting will involve some combination of ideas, but understanding each in isolation is worthwhile. In this work, we limit our scope to reducing forgetting by focusing on interference from the representation.
> >
> > > “The cited paper [1] uses the term “interference” for an inner product between gradient estimates evaluated at different data points and seeks, in contrast, to maximize interference.”
> >
> > That is one of the multiple different interference measures used in [1]. In Section 4 of our paper, we show that our interference measure is approximately proportional to negative gradient alignment, which is one of the interference measures used in [1] (e.g., see Section 5.2 of [1]).

---

> > > ### Comment · AnonReviewer1 · 2020-11-24
> > > **Response to authors' comments**
> > >
> > > We thank the authors for writing a detailed response.
> > >
> > > > We are not sure what the reviewer meant by "more powerful optimization method."
> > >
> > > Algorithm 4 in the Appendix describes a procedure similar to MAML with inner and outer loops updates of parameters $w_t$. This means that the method, like MAML, uses 2nd order derivatives for updating $w_t$ (even though they don't appear explicitly in the equations from the algorithm).
> > >
> > > > ...so we think it is unnecessary to compare to expensive methods introducing further approximation.
> > >
> > > Even if the said methods appear to be worse, the quality of the paper would be improved if a comparison with any other method that addresses interference and / or forgetting was added.
> > >
> > > > Even if the findings are unsurprising, they have not been shown by prior work, to the best of our knowledge.
> > >
> > > We believe that under mild assumptions, it should be possible to prove that the sensitivity of the output of a neural network to perturbations in a first layer parameter should be smaller than to the same perturbation in a last layer parameter. Since the result is going to be true in general, it is going to be applicable in the RL context as well.
> > >
> > > Based on the authors' comments, we increase the score to 5.

---

### Decision · Program_Chairs · 2021-01-07
**Final Decision**

**Decision:**

Reject

**Comment:**

The paper investigates interference in reinforcement learning and introduces a novel measure that can be used in value-based methods.
Although the reviewers acknowledge that the paper has merits (the topic is relevant and the paper is well written), they feel that the contribution is not sufficiently supported by either a theoretical or empirical analysis. The authors' responses have solved some of the reviewers' concerns, but they agree that this paper is not ready for publication in its current form.
I encourage the authors to update their paper following the reviewers' suggestions, in particular by improving the empirical analysis where comparisons with alternative methods (e.g., AVI/API methods that introduce regularization) need to be added.